# Supplementary Cementitious Materials in Building Blocks—Diagnosing Opportunities in Sub-Saharan Africa

Raine Isaksson [1,*], Max Rosvall [1], Arezou Babaahmadi [2], Apollo Buregyeya [3], Amrita Hazarika [2], Joseph Mwiti Marangu [4], Kolawole Olonade [5], Swaminathan Ramanathan [1], Anthony Rucukye [3] and Luca Valentini [6]

1   Department of Civil and Industrial Engineering, Uppsala University, 75237 Uppsala, Sweden; max.rosvall@angstrom.uu.se (M.R.); swaminathan.ramanathan@angstrom.uu.se (S.R.)
2   Department of Architecture and Civil Engineering, Chalmers University, 41296 Gothenburg, Sweden; arezou.ahmadi@chalmers.se (A.B.); amrita.hazarika@chalmers.se (A.H.)
3   Department of Civil & Environmental Engineering, Makerere University, Kampala P.O. Box 7062, Uganda; apollo@apollo.co.ug (A.B.); anthony.rucukye@mak.ac.ug (A.R.)
4   Department of Physical Sciences, Meru University of Science & Technology, Meru 972-60200, Kenya; jmarangu@must.ac.ke
5   Department of Civil and Environmental Engineering, University of Lagos, Akoka, Lagos 101017, Nigeria; kolonade@unilag.edu.ng
6   Department of Geosciences, University of Padua, 35131 Padua, Italy; luca.valentini@unipd.it
*   Correspondence: raine.isaksson@angstrom.uu.se; Tel.: +46-702490979

**Abstract:** Sustainable building should at least be affordable and carbon neutral. Sub-Saharan Africa (SSA) is a region struggling with housing affordability. Residential buildings are often constructed using block-based materials. These are increasingly produced using ordinary Portland cement (PC), which has a high carbon footprint. Using alternative Supplementary Cementitious Materials (SCMs) for block production might reduce the footprint and price. The purpose is to assess the level of information for SCM use in blocks in SSA and to use this information for Diagnosing the improvement potential as part of an Opportunity Study. Results from the scoping review show that aggregated information on SCMs and the quantities available is limited. Diagnosing the theoretical improvement potential in using cassava peel ash, rice husk ash, corn cob ash, volcanic ash and calcined clays, indicates that SCMs could represent a yearly value of approximately USD 400 million, which could be transferred from buying cement to local production. The use of SCMs could save 1.7 million tonnes of $CO_2$ per year and create some 50,000 jobs. About 5% of the PC used for block production could be substituted, indicating that, in addition to using SCMs, other solutions are needed to secure production of sustainable blocks.

**Keywords:** sustainable housing; supplementary cementitious material; sustainability opportunity study; diagnosing potential; alternative binders; block production; sub-Saharan Africa

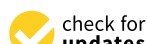



## 1. Introduction

According to the UN-Habitat and the World Bank, the demand for new affordable and sustainable housing is estimated at 300 million units for 3 billion people by the end of 2030 [1]. Sub-Saharan Africa (SSA) is a poor part of the world with 34 out of 49 countries classified as Least Developed Countries (LDCs). The current SSA population is about 1.1 billion people with a rapid population growth. With poverty, comes poor housing and the current situation requires substantial improvement. The rapid economic development of SSA further increases demand for housing and infrastructure. This rapid growth has created significant hurdles for urban and infrastructure planners, as well as the entire building industry [2]. Residential building is an important area of priority in most countries, but especially in low-income countries with young populations. Cement is needed to provide the needed residential building development.

Current yearly cement consumption for SSA is estimated to about 130 kg per person with a growth of 6–10% annually [3]. For residential buildings, a common solution is a block-based material that enables intermittent small investments and incremental construction of residences over a long duration [4]. Cement-based blocks are common and in many areas, their use is increasing due to their versatility and ease of use. The binder used in these blocks is commonly Portland cement (PC) produced in a modern cement plant. Cement drives the carbon footprint of the building blocks. In low-income countries cement also drives the price of the cement application, in the case of SSA, the block. Since cement prices in SSA are similar or higher than in developed countries this means that cement prices relative to salary are much higher in SSA. The main reason for high cement prices is mostly high costs of energy and high costs for long transport distances, considering that Portland cement is both an energy-intensive and a bulky material.

Cement and sand blocks, or sandcrete blocks, are earth dry masses with zero slump, with low cement content and low compressive strength. These blocks are often produced close to where they are used in small and simple production units. Typically, there is a mixer and a vibrating unit for compaction. Since strength requirements are low, these blocks could be good choices for using alternatives to PC.

Locally produced SCMs could be used to reduce the need of PC in block production. It might also be possible to fully substitute the PC by preparing a mix of SCMs with slaked lime, $Ca(OH)_2$. This would be similar to how cement was produced in Roman times.

Despite seemingly good availability of Supplementary Cementitious Materials (SCMs), such as agricultural waste ashes, calcined clays and natural pozzolanic materials [5–8], we have not found information on the use of these materials in concrete production or in cement block production in Africa. Volcanic ash is used as cement raw material in countries such as Kenya and Uganda, but not directly in block production. The effect on cement price from the use of volcanic ash in cement production seems to be limited. Cement prices still remain high, even if cement plants are using volcanic ash to substitute cement clinker. Possibly the price of a binder based on volcanic ash could be reduced by producing it in a smaller industrial unit with a small-scale cement mill.

A global perspective on the use of SCMs and fillers is presented by [6], where calcined clay, filler, fly ash, slag, natural pozzolan, vegetable ashes, waste glass and silica fume are compared, in terms of annual availability, with the consumption of PC. The implication from the study [6] is that globally calcined clay and fillers are the major alternatives for PC substitution, followed by fly ash and slag. For the African context, fly ash and slag have limited relevance since these materials are generally not available. Fillers are not considered since these are not pozzolanic, but inert materials.

In SSA there might be greater potential for natural pozzolans and vegetable ashes. The production of cement-based blocks in SSA relies on basic labour-intensive technology, often deployed close to the place of use of the blocks. PC is produced in most of the SSA countries in a few large production units with typical capacities of 500 thousand to a few million tonnes per year. There is also some cement and cement clinker importation. Cement often needs to be transported for long distances, making it more expensive in rural areas. Therefore, rural areas with suitable agricultural production or areas with natural pozzolanic materials could be those that could benefit most from using SCMs to reduce or fully replace PC in building blocks. Using residual resources from agricultural processes could also increase resource efficiency, avoiding disposal of waste in the local environment while creating employment.

This paper examines the opportunities for SCMs substitution of PC in building block production with focus on residential buildings in SSA. Focus is on cement and sandcrete-blocks. The studied blocks could have some additions of gravel or aggregates. They are typically produced in simple production units, compacted, demoulded, cured with water and stored. They must have zero slump to avoid deformation after casting. These blocks are produced in different sizes, and they could be solid or hollow. The solid blocks are those with the lowest strength requirements and would therefore be best suited for production

with alternative binders. Soil cement blocks, which consist of clay rich materials, which are stabilised with cement, could be an interesting area of research but these are excluded in the presented assessments and remain a future area for research.

Requirements for sandcrete blocks in terms of compressive strength are well below those of concrete and the minimum compressive strength required in many countries is about 2.2–3.5 MPa compared to some 18–35 MPa for concrete [9]. The strength requirements for blocks in residential buildings are relatively low. The load carrying capacity is a function of both the strength and the size of the block. By using larger blocks, making the load-carrying wall wider, the minimum compressive block strength, due to load bearing requirements becomes lower. The minimum compressive strength, described in block standards should mainly relate to durability. The wall-carrying load will be decided by the wall design, based on block dimensions and block compressive strength.

Since cement drives the costs and the price of the blocks, producers minimise its addition. Typical cement content by weight in the market segment selling to private users is often down to 5%. In comparison, low-strength concrete would have about 10% cement by weight.

Diagnosing the improvement potential requires that relevant data are available. Diagnosing is the first step in an opportunity study (DAS) [10]. The DAS consists of Diagnosing the improvement potential, Analysing the causes for the potential, and then Solving. When analysing the causes for the existing potential, one of the reasons for the identified improvement potential is often lack of data and lack of an adequate measurement system [10]. This suggests that one of the first steps should be analysing the maturity of the measurement system.

Diagnosing is performed with the purpose of assessing an approximate improvement potential, which will be used to decide if what is found justifies further work. A first iteration of diagnosing can be carried out to assess the approximate magnitude of improvement potential, based on limited data and including some assumptions. If the first diagnosing shows a substantial improvement potential, then there will be a reason to progress with better data and a more thorough analysis.

The main purpose of this study is to perform Diagnosing, which is the first step in an opportunity study. The work focuses on assessing the magnitude of the improvement potential in terms of the maximal theoretical substitution of PC in block production in SSA. Another purpose is to assess what information is available for SCMs accessible in SSA and their use in block applications. This is part of the second step of Analysing in the opportunity study. The quality of the measurement system has an important impact on the possibility to realise a detected improvement potential. This relates to the adage of: "What we cannot measure, we cannot improve". The improvement potential in PC substitution could be detailed in cost saving, carbon footprint reductions, and in increasing employment opportunities. The work is conducted by working with the research questions (RQs) of:

RQ1: What level of information is available in reviews of SCMs for block and concrete applications in SSA?

RQ2: What is the improvement potential in terms of affordability, $CO_2$ emissions and labour hours for SCMs in block making in SSA?

Conclusions are that the measurement maturity level for SCM use in blocks is low. Despite the limited information available, the theoretical potential for substituting Portland cement in block applications could be assessed and was found to be substantial. The SCMs could represent a maximal theoretical yearly value of some USD 400 million. The use of SCMs in blocks could save up to 1.7 million tonnes of $CO_2$ per year and create some 50,000 yearly jobs. About 5% of the PC used for block production could be substituted, indicating that in addition to using SCMs, other solutions are needed to secure production of sustainable blocks.

## 2. Theoretical Background

With research on SCMs and the application of them going back decades, there is an abundance of literature to cover. In this section, the theoretical methodology for identifying a sustainability opportunity is briefly covered together with a reasoning about the type of information and data that would be relevant to search for in the plethora of articles covering SCMs.

### 2.1. Sustainability Opportunity Study for SCMs in SSA

A Sustainability Opportunity Study (SOS) is a further development of an opportunity study [10,11]. An opportunity study consists of the three steps of diagnosing, analysing and solving and results in either discarding or accepting a proposal for change to obtain some improvement. Diagnosing requires that there are agreed performance indicators, data for performance and a goal. This enables the assessment of an improvement potential as the difference between a goal or assessed best possible process performance and current performance. If the identified improvement potential in the studied process has sufficient value, then the causes are analysed. If the causes can be identified, then possible solutions are discussed. If feasible solutions are found for the studied process, an opportunity of improvement can be presented to management. In this current work there is no clear identifiable management. Instead, there are several stakeholders with interest in managing resources sustainably. Two of them are the research collective and businesses in sub-Saharan Africa (SSA) interested in pursuing opportunities with SCMs. This study only covers the part of diagnosing, presenting an approximate improvement potential and partial analysing with focus on assessing the maturity and relevance of the measurement system.

We apply a process approach, which enables studying performance at different levels. The process could be within a company, an organization, the entire value chain that an organization is part of, a regional process, such as providing block binders in SSA, or a global process of, e.g., cement production [12]. Performance is expressed using value and harm indicators.

When discussing sustainability, it often could be difficult to agree upon what exactly sustainability is and how it should be measured. It seems to be difficult to agree on definitions for, e.g., sustainable building, sustainable education, and sustainable tourism [11]. This could be a general problem, which has serious consequences, since when we do not agree on where we are and where we should go; there will be little sustainability improvement. Isaksson and Hallencreutz [13] suggest that leading change requires that it can be communicated, which requires measurements and agreed definitions based on a common understanding of the process in focus. Without agreed KPI the diagnosing of an improvement potential cannot be carried out. The solution is, to start with Understanding, Defining and Measuring the performance in a chosen process, in what is called a Sustainability Opportunity Study [11]. The matrix in Table 1 combines the three steps of the Opportunity Study with the three first stages of Understanding–Defining–Measuring–Communicating–Leading Change [13] and describes the part of the Sustainability Opportunity Study which has been done.

**Table 1.** Matrix for combining the opportunity study steps DAS with the first three stages of UDMCL.

|  | Understanding | Defining | Measuring |
|---|---|---|---|
| Diagnosing | The studied process of providing SCMs for blocks in SSA | | |
| Analysing | Only analysing the resource of the measurement system | | |
| Solving | | | |

In Table 2 details of Diagnosing, Defining and Measuring are described. The SOS is needed when clear and logically justified KPIs for a process are missing. Good sustainability KPI rely on a clear and relevant sustainability definition. This requires a common understanding of sustainability in the studied system.

**Table 2.** The three first steps of the Sustainability Opportunity Study (SOS). Original table courtesy of Isaksson et al. (2022) with permission from TQM Journal [11].

| Understanding | Defining | Measuring |
| --- | --- | --- |
| Scope, using value chain from cradle to grave by defining input, output and business model. Identifying main sustainability stakeholders, their value needs, and the harms, they are subjected to in the value chain with focus on climate, biodiversity, and poverty as well as any other significant harm as identified. Defining the qualitative improvement potential as the difference between possible and/or required performance and current performance. | Based on the Pareto principle define the vital few stakeholders, value needs and harms caused. Focus on people and planet needs and convert this to proposed definitions for sustainability and sustainable development, which can be operationalised. | Measure sustainability as a state and sustainable development as change. Identify value and harm indicators—the KPIs (y-values) that can be used to describe current sustainability and the sustainability performance over time. Value and harm are expressed in terms of impacts on people, planet and profit KPIs should be expressed in absolute and relative terms. Assess the quantitative improvement potential for chosen y-values in terms of level and rate of change. |

Understanding sustainability should include understanding that an organisation needs to work with sustainability impacts in the entire value chain from cradle to grave and that the main stakeholder impacts should be understood in it [11]. By focusing on the vital few impacts in the value chain, such as climate change, loss of biodiversity and poverty it becomes possible to propose a definition. Sustainability should be seen in terms of both generating value and causing harm [11].

Isaksson et al. [11] have in Table 3 applied the logic of Diagnosing, Analysing, Solving on the residential building value chain, proposing a sustainability definition and KPIs. Sustainable building is defined as at least affordable and climate neutral. This reflects the human right of shelter which relates to poverty and the climate effect which is important in the building value chain. The assessment of the improvement potential is based on Back-casting [14], where a visionary state has been compared with a current performance. The main impacts identified under Understanding are providing shelter and effects on climate.

**Table 3.** Visualising diagnosing for the value chain of building, based on Understanding, Defining, Measuring of Diagnosing (UDM-D). Original table courtesy of Isaksson et al. (2022) with permission from TQM Journal [11].

| Value Chain of | Understanding | Defining | Measuring (Value/Harm) | Summary Improvement Potential |
| --- | --- | --- | --- | --- |
| Residential Building | Main value is affordable shelter and main harm is climate effect | At least affordable with zero-carbon footprint | Living space per price and carbon footprint | 8 Gtonnes of $CO_2$/year. Huge deficit in appropriate housing |

Based on values from Statista 2021 [15] the recorded yearly total global $CO_2$ emissions were 37 Gt. This year still suffered from the effects of Corona, implying that emissions probably will increase rapidly. We use an estimated value for the end of 2022 of approximately 40 Gt $CO_2$/year as a base. Some estimates put the building value chain carbon emissions when counted from cradle to grave as high as 40% of global carbon emissions or about 16 Gt/year [16]. Out of this, about 50% could be accounted for residential building or about 8 GT $CO_2$ per year. The social improvement potential is the aggregated lack of appropriate housing.

Carbon emissions from cement production total are by Statista reported to be 4.5% [17] but this is most likely on the low side since global cement production is estimated as 4.2 billion tonnes in 2020 [18] with a carbon content of about 590 kg per tonne cement [19], which results in about 6.7% of the global emissions. There is a certain risk of under reporting since high carbon emissions mostly are seen as a liability. Here we assume that about 7% of global emissions or about 2.8 Gtonnes $CO_2$/year originate from cement. The cement emissions constitute a major part of the concrete and material emissions. With approximately 8% coming from concrete, where most of this is from cement, the building materials $CO_2$-emissions would represent about 20% of the total building value chain emissions. With low energy houses and with increased use of renewable energy, the proportion of building material out of the total building value carbon emissions increases. With focus on affordable housing in sub-Saharan Africa, the focus on price of materials will be higher. This is since most low-cost houses will neither have cooling nor heating. This puts focus on the cement contribution to carbon emissions globally and particularly in SSA.

The housing deficit is serious in sub-Saharan Africa (SSA), one of the poorest regions in the world. About 1.1 billion people live in SSA but the population is predicted to double by 2050 and to reach 4 billion by 2100. Current cement consumption is low, with an approximate yearly rate of about 130 kg/person, compared to a global average of almost 600 kg/person. Isaksson and Buregyeya [20] estimate that, based on the predictions of population growth and assuming that appropriate shelter is provided, about 2 Gtonnes/year of cement would be needed in 2100 for SSA only. This would correspond to about half of the current global cement production. Using PC based on current practices would generate a huge carbon footprint. The global cement industry proposes Carbon Capture and Storage as the main solution for reducing carbon emissions [6]. This is an expensive process, which risks doubling the cement price with severe effects for poor people. Another solution for reducing the carbon footprint is to use Supplementary Cementitious Materials (SCM), which could be of particular interest in SSA.

*2.2. Theory of Use of SCMs*

SCMs are usually divided into natural and artificial pozzolans [5]. The pozzolanic property refers to the ability to form calcium silicate hydrates by reaction between the soluble silica fraction present in SCMs and Portlandite-Ca(OH)$_2$-formed during PC hydration [21]. Examples of natural pozzolans are volcanic ash and calcined clays and artificial pozzolans are referring to the materials, often by-products, obtained from human production processes. Artificial pozzolans and fillers include the following examples: saw dust ash [22–25], rice husk ash [22–28], oyster shells [23,27,29], sewage sludge ash [27], or ground glass [8,27]. The essential part of a good SCM is that the combined chemical composition of silica, alumina and iron exceeds 70% of the pozzolanic material [30]. More recently, specific guidelines in terms of $Al_2O_3$ and $SiO_2$ content were provided for the use of calcined clay as pozzolanic material [31].

Substantial research has been carried out to test the usability of agricultural waste products, such as rice husk ash, corn cob ash and cassava peel ash. The key for each variant of potential pozzolan is the ability to substitute the standard binding material, PC, without excessive loss in terms of compressive strength performance. Based on the logic in Section 2.1, identifying affordability and $CO_2$ emissions as key sustainability impacts, the pozzolan should be able to substitute the binding capacity of the PC binder. The scope for the value chain from cradle to grave for building is interpreted as extracting SCM materials followed by binder production, block production, building, use of buildings, demolition and reuse of materials. The value produced is m$^2$ housing but also employment. The main harms would be cost of materials and carbon footprint. For block production we delimit the process ending it with a block having been placed in a residential wall. The wall functionality should be acceptable and safe. At this stage standard requirements are not included since these vary between countries and their relevance could be discussed. Generally, blocks with strengths in the range 1–7 MPa are used. The key issue is the block

durability. The required wall carrying capacity can be achieved with low-level strength by using thicker blocks. The value produced is an m$^2$ of acceptable wall. The main harms are the cost of the m$^2$ wall and the carbon footprint. Additionally, a value produced is the hours of employment the production and use of pozzolans has created. The expectation is that this substitution leads to a lower carbon footprint and lower price. For carrying out diagnosing, the potential of SCM sustainability opportunities in SSA, figures for material availability, means for preparation of the material to achieve pozzolanic abilities, and material strength performance are needed. With information of the quantity of SCMs and their cement substitution potential, we can make a first assessment of how much cement we could substitute. We can compare this to the total cement consumption, and we can estimate the maximal value that SCMs could produce.

### 3. Methodology and Data

The RQ1 about the level of information available for use of SCMs in blocks and concrete is answered by conducting a scoping review and then analysing the data using a proposed checklist presented below.

This opportunity study uses the methodology of reverse engineering [32] starting with the need for sustainable housing in the SSA region. For the purpose a checklist that summarizes the information needed has been developed. This specifies what a measurement system should include to provide necessary information for diagnosing an improvement potential:

Information about availability in a chosen region:

- Natural pozzolan: amount of available reserve of material, e.g., the size of the cover of volcanic ash from previous eruptions.
- Artificial pozzolan: Production of residual material that can be converted into a pozzolanic material. For example, amount of corn cob ash that can be obtained by incineration of cobs during production of corn. Information about means of preparation of the pozzolanic materials, such as calcining and grinding.
- Natural pozzolan: what processing is required to obtain a reactive pozzolanic material?
- Artificial pozzolan: what processing is required to obtain a reactive pozzolanic material from the residual material?

Information about the usability:

- Natural and artificial pozzolan: The percentage of the pozzolanic material that can substitute PC while retaining the performance in terms of compressive strength. This could be generalized into a performance indicator for chosen SCMs for a fixed recipe.
- Natural and artificial pozzolan: the specific application for which performance tests have been conducted, e.g., standard concrete (10+ MPa) or low-strength applications (1–5 MPa).

Using the checklist, we can collect and analyse the essential information needed to evaluate the opportunities for housing in SSA and we can assess the level of information. For examining the available information about potential in SCMs for SSA, several data sources have been used based on the character of each guiding research question. RQ1 has been answered through a scoping review, for the purpose of finding an overview of the main characteristics of the current available research on performance of alternative binders [33]. The scoping review is 1 out of 14 different types of literature reviews described in [34] and it is a suitable method for synthesising research evidence and mapping literature in a given field in terms of its nature, feature, and volume [35]. For RQ2, some information lacking from the ten review articles was complemented by input from experts, as wells as co-authors of this paper. Beyond this, desk research was conducted to fill gaps with available information regarding cement consumption, block production and main alternative binders. When information has not been accessible, estimations have been used to complete the diagnosing of the improvement potential.

The scoping review was performed according to guidance from [33] following the four steps of identification, screening, eligibility and inclusion. Identification was conducted through an initial search in the Scopus database using the search string for title, abstract and keywords: ((cementitious AND supplementary AND material) OR (alternative AND binders) AND (sub-Saharan AND Africa)). This resulted in five hits where only one was available to the authors in full-text, but later dismissed due to lack of relevance for the purpose of identifying review articles. Further identification was performed in the Google Scholar database using the function for selection of review articles. Here, combinations of the following keywords were used to identify titles of potentially relevant review articles:

- Supplementary Cementitious Materials
- Alternative Binders
- Concrete
- Brick or Block
- Review
- Low strength
- Sub-Saharan Africa

See Table 4 for five combinations used for identification in the Google Scholar database. The resulting identification from searches in Google Scholar included 432, where the article found in Scopus also was included. The screening was performed on titles where inclusion criteria was based on inclusion of the above listed keywords. The screening resulted in a list of 10 articles, of which abstracts were further screened for eligibility. Criteria for eligibility were being a review of SCMs mapping or comparing different alternatives to Portland cement, and all articles were included. The resulting articles are presented in Table 5.

To increase the rigorousness of the scoping review, co-authors of this paper, representing experts on SCMs, were invited to provide literature on research into SCMs that they have been conducting or that they have knowledge about. From the results from the scoping review combined with the expert researcher's input provided a short list of SCM, serving as a limitation to the part of diagnosing in an opportunity study. For deriving potential opportunities for improvement in sustainability performance of blocks in buildings, the selection criteria were availability in SSA. Availability was here defined as the SCM being a significant resource in terms of amounts and the production processes taking place in SSA. This resulted in the SCM included in the study presented in Table 6.

**Table 4.** Description of article identification.

| | Search 1 | Search 2 | Search 3 | Search 4 | Search 5 |
|---|---|---|---|---|---|
| Include in title, abs, text | cementitious supplementary material alternative binders review Africa | cementitious supplementary material alternative binders review Africa concrete | cementitious supplementary material alternative binders review low strength concrete Africa | alternative binders review low strength concrete Africa | review low strength concrete Africa |
| Must include exact phrase | low strength concrete | low strength | | cementitious supplementary material | alternative binders |
| Include one of the two | block brick | block brick | block brick | block brick | block brick |
| Choice | review articles | review articles | review articles | review articles | review articles |
| Results | 13 | 69 | 330 | 0 | 20 |

**Table 5.** Selected review articles from scoping review.

| Title | Year of Publication | Journal | References |
|---|---|---|---|
| Agricultural wastes as aggregate in concrete mixtures—A review | 2014 | Construction and Building Materials | [26] |
| Supplementary cementitious materials origin from agricultural wastes—A review | 2015 | Construction and Building Materials | [22] |
| Green concrete partially comprised of farming waste residues: a review | 2016 | Journal of Cleaner Production | [29] |
| A review of waste products utilized as supplements to Portland cement in concrete | 2016 | Journal of Cleaner Production | [27] |
| Concrete using agro-waste as fine aggregate for sustainable built environment—A review | 2016 | Journal of Sustainable Built Environment | [23] |
| A huge number of artificial waste material can be supplementary cementitious material (SCM) for concrete production—a review part II | 2017 | Journal of Cleaner Production | [24] |
| Agricultural Solid Waste as Source of Supplementary Cementitious Materials in Developing Countries | 2019 | Materials | [28] |
| High volume Portland cement replacement: A review | 2020 | Construction and Building Materials | [8] |
| Incorporation of agricultural residues as partial substitution for cement in concrete and mortar—A review | 2020 | Journal of Building Engineering | [25] |
| Biomass ashes from agricultural wastes as supplementary cementitious materials or aggregate replacement in cement/geopolymer concrete: A comprehensive review | 2021 | Journal of Building Engineering | [7] |

**Table 6.** SCMs included in this study.

| Supplementary Cementitious Material | Abbreviation |
|---|---|
| Volcanic Ash | VA |
| Calcined Clay | CC |
| Rice Husk Ash | RHA |
| Cassava Peel Ash | CPA |
| Corn Cob Ash | CCA |

## 4. Results from Scoping Review on SCMs

The identified review articles provide an overview of what information is available in terms of the opportunities for substituting PC with SCMs. The results are presented in Table 7, where each pozzolanic SCM is reviewed based on the developed checklist criteria. The review articles further include fly ash, ground granulated blast furnace slag and silica fume, which all were excluded from the result table based on the low production capacity of these artificial pozzolans in SSA. The results show that availability in terms of production capacity in tonnes or hectares and alternative use of the residual materials are given for some, but not all, SCMs. Means of preparation from residual material to usable pozzolanic material is most often reported for, but with varying level of details. For example, the issue of grinding the ash from bio-residuals is seldom mentioned, even though it is indicated

that the processing increases the reactivity [25,28]. The usability in terms of classification as a pozzolanic material, with mapping of chemical composition, is frequently reported for all but four SCMs. The most underreported information is the rate of substitution of PC for the specific SCM. Several reviews include the testing of compressive strength for varying percentage of PC substitution. However, there are not tests indicating if the SCM can replace the binding capacity of PC, and how much of the SCM is needed to acquire the same compressive strength as the control mix with PC. Further, all tests reported in the reviews are conducted with more than 10% PC binder, as commonly used in ordinary concrete, but not in block production, where cement content often is about 5%. In general, the full information about the test mixes is missing for several of the review articles.

The ten review papers collectively reported for 33 SCMs, out of which 23 were included in the SCM overview, see Table 7. Fly ash, silica fume and ground granulated blast furnace slag were excluded based on the lack of established industry that produces those types of by-products locally in SSA [36]. Further, coconut shell, tobacco, sisal, cork, and date palm were excluded based on their use as aggregates and not binders in concrete applications.

**Table 7.** Results from scoping review of SCM.

| SCM and All Review References That Cover the SCM | Availability— Reserves and Resources | Availability— Alternative Use | Means of Preparation | Usability— Content of Pozzolanic Components (SiO$_2$, Al$_2$O$_3$, ... ) | Usability— Rate of Substitution of PC | Usability— Tested for High (>10% Binder) or Low (<10% Binder) Strength Performance |
|---|---|---|---|---|---|---|
| Corn Cob Ash | Tonnes and Hectares [7,22,24] | Waste product or feedstock for biogas production, ash disposed of to landfill [7] | Burning waste products in 550 °C [7], 650 °C [25], 700+ °C [24,29] | [7,22,24–26,29] | Not Available (N/A) | High [7,22,24–26] |
| Rice Husk Ash | Tonnes [22,24,26,27] | Animal feed, fire making, litter material, making concrete, board production, reinforcing ceramic cutting tools, but mainly disposed waste [22,23,26] | Waste from husking process of rice, about 20% of rice production is husk, burning produces ash [22,24–27] | [22,24,25,27] | N/A | High [22,24–26] |
| Saw Dust/Wood Waste Ash | N/A | Disposed in nature as waste [22,24] | Waste by-product from various wood production and combustion of residuals [22,23] incinerating at 650 °C to produce ash [25] | [25] | N/A | N/A |

**Table 7.** *Cont.*

| SCM and All Review References That Cover the SCM | Availability—Reserves and Resources | Availability—Alternative Use | Means of Preparation | Usability—Content of Pozzolanic Components (SiO$_2$, Al$_2$O$_3$, … ) | Usability—Rate of Substitution of PC | Usability—Tested for High (>10% Binder) or Low (<10% Binder) Strength Performance |
|---|---|---|---|---|---|---|
| Sugarcane Bagasse Ash | Tonnes [23,27,28] | Disposed in nature as waste [24], landfill [28] | Waste from crushing of sugar cane, turned into ash through combustion/cogeneration at 600–650 °C [22] often used for boiler fuel [23,24,27], the reactivity can be increased through milling or grinding the ash [25,28] | [22–25,28] | N/A | High [22–24] |
| Palm oil fuel ash | Tonnes [24,25,27] | N/A | Waste by-product from bio-diesel industry, incinerated through combustion to ash [24,27], the reactivity can be increased through milling or grinding the ash [8,25,28] | [8,25,27,28] | N/A | High [24,28] |
| Bamboo Leaf ash | Tonnes [7,28] | Disposed in nature as waste [7,29] | Waste by-product from bamboo agriculture, combustion in electric furnace at 600 °C produces ash [7,22,24,29] | [7,28,29] | N/A | High [28] |
| Wheat Straw Ash | Tonnes [7,29] | Disposed in nature as waste [7,29] | Waste by-product from wheat agriculture, combustion in electric furnace at 570–670 °C produces ash [7,29] | [7,29] | N/A | N/A |

**Table 7.** *Cont.*

| SCM and All Review References That Cover the SCM | Availability—Reserves and Resources | Availability—Alternative Use | Means of Preparation | Usability—Content of Pozzolanic Components ($SiO_2$, $Al_2O_3$, … ) | Usability—Rate of Substitution of PC | Usability—Tested for High (>10% Binder) or Low (<10% Binder) Strength Performance |
|---|---|---|---|---|---|---|
| Barley Straw Ash | N/A | N/A | Waste by-product from barley agriculture, combustion produces ash [37] | [7] | N/A | N/A |
| Olive Waste Ash | Hectares [29] | Disposed in nature as waste [29] | Waste by-product from olive agriculture, combustion in electric furnace at 600–800 °C produces ash [37,38] | [7,29] | N/A | N/A |
| Banana Leaf Ash | Tonnes [7,29] | N/A | Waste by-product from banana agriculture, combustion in electric furnace at 800-900 °C produces ash [29], milling can increase reactivity [7] | [7,29] | N/A | High [7] |
| Elephant Grass Ash | Ton/Ha [7] | Animal feed, charcoal production, bio-ethanol production [7] | Waste by-product from energy production through combustion, require pre-treatment before combustion [29] | [7,29] | N/A | N/A |
| Oyster Shell | Tonnes [29] | Disposed in landfill [29] | Residual waste from aquaculture [23,29], could be washed, burnt and milled [27] | [23,29] | 0—no binder effect [29] | High [23] |
| Periwinkle | N/A | N/A | Residual waste from aquaculture [29] | N/A | 0—no binder effect [29] | N/A |

**Table 7.** *Cont.*

| SCM and All Review References That Cover the SCM | Availability—Reserves and Resources | Availability—Alternative Use | Means of Preparation | Usability—Content of Pozzolanic Components ($SiO_2$, $Al_2O_3$, … ) | Usability—Rate of Substitution of PC | Usability—Tested for High (>10% Binder) or Low (<10% Binder) Strength Performance |
|---|---|---|---|---|---|---|
| Mussel | N/A | N/A | Residual waste from aquaculture [29] | N/A | 0—no binder effect [29] | N/A |
| Ground Glass | Tonnes [27] | Recycled into glass, or disposed in landfill [27] | Waste from consumption, needs sorting and milling [8,27] | [8,27] | N/A | N/A |
| Sewage Sludge Ash | Tonnes [27] | Fertilizers, bio-gas for fuel, fuel for incineration or dumped in landfill [27] | Residual waste from water management processing [27] | [27] | N/A | N/A |
| Groundnut Shell | N/A | N/A | Waste by-product from ground nut agriculture [23] | N/A | N/A | [23] |
| Wild Giant Reed Ash | N/A | N/A | Waste by-product from wild giant reed agriculture, combustion produces ash [23] | N/A | N/A | [23] |
| Ceramic Waste Powder | N/A | N/A | Waste by-product from ceramic production of bricks, tiles and other products [8] | [8] | N/A | N/A |
| Neem Seed Husk Ash | N/A | N/A | Waste by-product from neem oil production, incineration of husks produces ash [25] | [25] | N/A | High [25] |
| Rice Straw Ash | Tonnes [7] | Disposed as bio-waste on farmland [7] | Residue from rice harvest, incineration of straw produces ash [7] | [7] | N/A | N/A |

**Table 7.** *Cont.*

| SCM and All Review References That Cover the SCM | Availability—Reserves and Resources | Availability—Alternative Use | Means of Preparation | Usability—Content of Pozzolanic Components ($SiO_2$, $Al_2O_3$, ...) | Usability—Rate of Substitution of PC | Usability—Tested for High (>10% Binder) or Low (<10% Binder) Strength Performance |
|---|---|---|---|---|---|---|
| Corn Stalk Ash | N/A | N/A | Waste by-product from corn production, dried stalks are incinerated at 600 °C [25] | [25] | N/A | N/A |
| Corn Husk Ash | N/A | N/A | Waste by-product from corn production, dried stalks are incinerated at 600 °C [25] | [25] | N/A | High [25] |

## 5. Sustainability Opportunities for Block Production in SSA

Following the steps for diagnosing, this section builds on the understanding from the scoping review presented in Section 4, and complementary input from researchers in the field and desk research. With the identified KPIs as cost, $CO_2$ emissions, affordability and labour hours created from use of PC in block production, Section 5.1 provides the background to estimate the total use of PC for block production. This is followed by Section 5.2, where the estimation for calculated cost and $CO_2$ emissions for housing needs is described. Section 5.3 discusses a theoretical target performance, Section 5.4 provides the available information for the selected SCMs, and Section 5.5 presents the resulting opportunities for improvement in terms of sustainability performance.

### 5.1. Assessment of Current Block Production in SSA

Cement consumption in SSA has had a continuous growth of 6–10% per year in the period 2012–2020. A combination of growth in population and consumption of cement per person drives the increasing cement consumption (see trends in Figure 1). The average per capita cement is assessed to be about 131 kg cement/capita in 2021, based on an extrapolation of Figure 1. Based on the same data, the population in SSA is assessed to be about 1.1 billion in 2021. This, combined, corresponds to a total cement consumption in SSA of 144 Mtonnes in 2021. For calculating the improvement potential 150 Mtonnes of PC per year in SSA has been used.

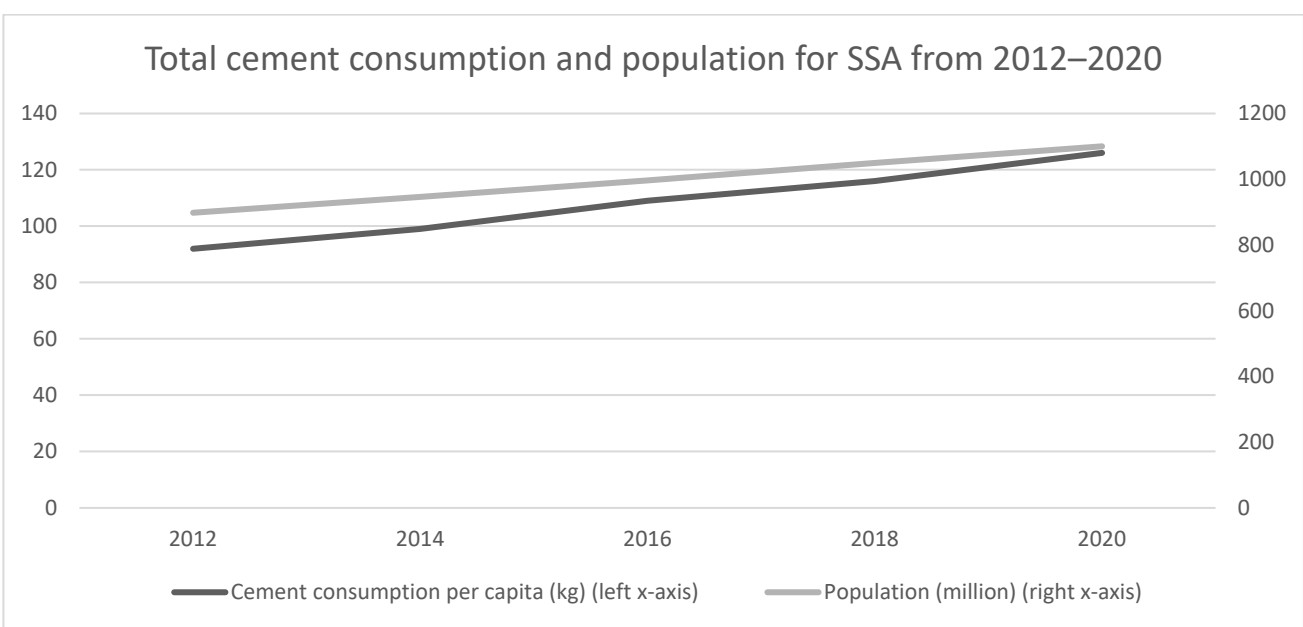

**Figure 1.** Trends in cement consumption per capita and population over the period of 2012–2020 in SSA.

Data from mapping of housing in SSA, during the period 2000–2015, indicate that cement is one of the major materials used for walls construction [4]. Out of 96 included national surveys from countries in SSA during 1991–2015, 62 provided information about materials used for finished walls [4]. The classifications that included the use of cement were: concrete (included in two surveys), stone with lime or cement (46), cement block (55), cement (47) and burnt bricks with cement (5), resulting in 155 classifications using cement as part of building material, out of the totalling 274 classifications used in the 62 surveys (see [4] for more details). It is further reported that construction of new houses represents the larger share (70%) of cement consumption in emerging markets, to be compared with about 30% in Europe and 15% in the US [39]. Estimates for the local market in Dar es Salam indicate that 70% of the cement consumption is used for sandcrete block production, i.e., low-strength blocks using PC as binder [40,41]. In order to establish some type of reference for cement-based block production we use the figure of 70% of the cement used for residential buildings. Out of this, 60% is for blocks in walls. This corresponds to about 63 million tonnes of cement for blocks per year in SSA (150 Mt/year × 0.7 (for residential housing) × 0.6 (for blocks in walls). A typical cement addition by weight is about 5%, which would mean that about 1260 million tonnes of blocks are produced per year. Using a reference block from Tanzania, the 6∗9∗18-inch solid block, which weighs about 30 kg, the number of blocks counted as 6-inch blocks becomes about 42 billion blocks. This indicates the magnitude of the market opportunity for low-strength binders. This figure can also be used to calculate the need of labour hours.

*5.2. Cost and Carbon Footprint as Function of Cement Content in Blocks*

When assessing the value of residential buildings one important value criterion is available space expressed in m². Apartment space is an important functional value that can be compared to price and the carbon footprint. A further simplification of the functional value is to use m² wall as the functional value. In a basic house, some 60% of the material used for the house goes to walls. Isaksson and Buregyeya [20] use m² of wall as the value with which price and carbon emissions are compared. For comparing how the choice of binders affects costs, carbon footprint and working hours, m² of wall is a suitable reference.

Block dimensions affect the number of blocks needed for an m² wall. Wider blocks result in more material in the m² of wall and build a stronger wall. However, increasing

wall thickness and compressive strength of blocks beyond what is needed does not increase the user needs value. This means that walls could have different thickness but the same functional value.

The price is calculated based on the number of blocks needed for a m$^2$ wall and the carbon footprint is calculated based on the quantity of cement used in the blocks. Cement content in blocks is often minimised since cement is the most expensive ingredient. Typical cement content by weight in sand cement blocks is 5–7% by weight. This results in a 28-day compressive strength of 1–7 MPa. Most block standards in SSA require 2.2–3.5 MPa [9], but this value is presumably needed to guarantee durability. Simple calculations show that a 1 MPa block, which is 6 inches (15 cm) wide, 9 inches tall (23 cm) and 18 inches (45 cm) long can carry the weight of six tonnes [40]. From a functional point of view, this is more than enough. People buying and producing blocks for residential buildings seldom control strength requirements in terms of measured compressive strength since this would mean a significant extra cost. Instead, the technicians dealing with the construction have developed an understanding for what is good enough by feeling, scratching and hitting the blocks [40]. Based on this reasoning, we carry out assessments for the improvement potential using 5% cement for blocks that, depending on production practices, will result in strengths of about 1–5 MPa at 28 days.

Cement carbon emissions vary depending on the cement clinker content and the clinker carbon footprint. In block production it is common to use cements with limestone that have a lower clinker % but often also a poor strength performance. The carbon footprint is calculated using a reference from Isaksson and Babatunde [42] that suggest 700 kg CO$_2$/tonne of cement.

*5.3. Block and House Sustainability Performance*

The reference performance is calculated using 6-inch solid blocks with the dimensions 6∗9∗18 inches and refers to results from Isaksson and Buregyeya [20], but also on experiences from block production in Tanzania producing solid 6-inch sandcrete blocks [15,40]. Mixes in Tanzania are based on number of blocks per 50 kg bag of cement. A common mix produces about 28–34 pieces of 6-inch blocks per bag of cement. These blocks, when appropriately compacted, weigh about 30 kg. With 5% of cement the number of blocks per 50 kg bag is 33. The number of 6-inch blocks needed for 1 m$^2$ is taken from Isaksson and Buregyeya [20] and set as 9.1/m$^2$. Here, joints have been set to zero, which is a simplification. However, by using the same number of blocks when assessing blocks with SCMs this still allows for a valid comparison of binder effect. Cement prices vary and are generally higher in SSA than in the rest of the world. Cement prices are higher in rural areas, with prices going above USD 200 per tonne. Here, a cement price of USD 150/tonne is used or USD 7.5 per 50 kg bag. For assessing the number of houses, we propose a standard one-storey house of 96 m$^2$ living area. We use a wall height of 3 m and assume a house of 16 times 6 m, which results in an outer wall length of 44 m. We add 26 m for internal walls, which results in a total wall length of 70 m. This results in 210 m$^2$ of wall, which should be reduced by some 10 m$^2$ for windows. With some losses, this is assessed to result in the need of about 2000 blocks. The cement used for the walls is considered as 60% of the total need of cement. The number of persons staying in the 96 m$^2$ house is set to 6, which is in the interval of a typical household, based on assessment by the authors for Uganda (5–10 persons per household). As an indicative figure, this can be considered acceptable for comparison purposes.

Based on earlier calculations, 63 Mtonnes of cement per year are used for block production in walls. With the price of USD 150/tonne, the estimate is that residential buildings use about USD 9 billion yearly for blocks. This is an extreme simplification with the purpose to assess the magnitude of the improvement potential.

In Table 8, the reference results for a m$^2$ wall and for a standard house have been assessed based on the use of PC.

**Table 8.** Calculated cement consumption, cost, and carbon footprint for sandcrete blocks, blocks in $m^2$ wall, wall for house and for walls in SSA.

| Performance Criteria | For Block | For $m^2$ Wall | For House (6 Persons) Walls Only | For SSA Residential Building with Blocks Going to Walls |
|---|---|---|---|---|
| Cement amount | 1.5 kg | 14 kg | 2000 Blocks*1.5 kg/Bl. = 3 tonnes | 63 million tonnes |
| Cement cost | USD 0.23 | USD 2 | USD 450 | USD 9 billion |
| Carbon footprint | 1.1 kg | 9.6 kg | 2.1 tonnes | 44 million tonnes |

In Table 8, only blocks in walls are included. Blocks might also be used for other parts of the building, such as for the foundation, but these are not included in the estimate.

Calculating the employment effect in substituting PC with SCMs can only be performed at a superficial level. Block prices vary but are based on results from Tanzania at about 0.7 USD per 6-inch solid block. With 5% cement or 1.5 kg per block, the total number produced out of 63 Mt cement is about 42 billion blocks. The total sales value using USD 0.7 per block is USD 29 Billion. The assessed cost for work for producing the blocks is about 25% of the sales price, or USD 7 Billion. Monthly salaries vary and could be as low as USD 50/month. We use here a low urban salary, which would be USD 150/month or USD 1800/year. Based on this, the yearly block production creates about 4 million jobs in SSA. Currently, the cost of cement is twice the cost of labour. By using SCM, the cost of cement could be decreased, and the cost of labour increased, meaning that more employment could be created while maintaining the same cost. The economic improvement potential could either be calculated as cost saving or as increased employment.

### 5.4. Pozzolanic SCMs as Substitution to PC

With pozzolanic materials being used since the ancient Roman times [43], there have been extensive research efforts to explore their reactivity and inherent composition, e.g., [5,8,16]. The global market for binders based on pozzolanic materials is largely related to use as a partial replacement in PC. The hydrated PC provides the $Ca(OH)_2$ needed for the activation of the pozzolanic materials. The ancient Romans used burnt lime, which still today could be a cheaper solution, compared to cement. Even low grade limestone could be used for preparing burnt lime (CaO) and then slaking it with water to form $Ca(OH)_2$. For calculating the potential of pozzolanic materials we assume that either PC, Quick Lime (CaO) or slaked lime $Ca(OH)_2$ will be available to activate the pozzolanic materials. All the SCMs presented in this section are being reported as potential substitutions of PC as binder in concrete applications.

#### 5.4.1. Cassava Peel Ash (CPA)

The cassava plant has its origin from South America, from where it spread to different regions of the world today. This perennial crop is known with different names, such as manioc, tapioca and yucca. The tubers, which are part of the root system of the crop, are processed for various uses. Cassava now provides about 30% of worldwide production of roots and tubers, and is the staple crop of over 200 million people in Africa alone, with a production of 138 million tonnes in 2018 [44]. In the course of processing cassava, either for food or industrial usages, large amounts of peels (major by-product of cassava) are produced. According to Adesanya et al. [45], between 20–30% by weight of cassava tubers are peels. The norm in most of the processing centres is that cassava peels are heaped and burnt for no benefit in return to give room for next generation of peels to be produced. The conversion from cassava crop to CPA is estimated to 5% based on [6]. The potential of its ash as a pozzolanic material has been investigated by a number of researchers who have used it successfully as a partial or total replacement of Portland Limestone Cement (PLC) in the production of sustainable concrete [46–51]. For example, Kumator et al. [51] used 10% of CPA in place of PC, while Olonade et al. [50] used CPA as a replacement for cement

at 5, 10, 15 and 20%, respectively. In another effort by Olutaiwo and Adanikin [47] the percentage replacement of CPA varied from 0–10%. Taku et al. [46] carried out an extensive study on the optimization model for compressive strength of sandcrete blocks using CPA blended cement mortar as a binder and the content of CPA used was within the range of 0–30%. In other scholarly works, the content of CPA had been increased to as much as 100% as reported by Edeh et al. [52]. If this SCM is attributed 20% of cement strength, then the total potential cement replacement in SSA with CPA is about 1.4 million tonnes per year.

### 5.4.2. Corn Cob Ash (CCA)

Corn cob is an agricultural solid waste from maize and corn. According to the Food and Agriculture Organisation (FAO), maize surpassed rice in 2001 to become the second most produced crop worldwide [44]. Globally in 2019, 1.15 billion tonnes of maize were produced across the world, of which Africa produces 78.90 million tonnes [44,53]. Maize is primarily produced by small-scale farmers in sub-Saharan Africa for sustenance as part of diversified farming production [54]. Adesanya et al. [55] obtained corn cob ash by burning corn cobs at a calcining temperature of 650 °C for 8 h. The conversion rate from corn crop to CCA is estimated to be 5% based on [6]. Various other studies reported similar calcining temperatures [56–58] to maximise the content of amorphous silica in the corn cob ash, and thereby improving its reactivity. To make CCA, first maize cobs are broken down into small pieces, which improves combustibility and lowers carbon content, which impacts the pozzolanic characteristics. Adesenya et al. [55] reported that the ash of corn cob has an amount of $SiO_2$ of more than 65% and a combination of $Al_2O_3$ and $SiO_2$ of more than 70%, which as per ASTM standards [59], implies that the cob ash can be used as a pozzolan in blended cement concrete. They further report that at an optimum replacement level of 8% of PC with CCA results in a significantly improved later age compressive strength when used for standard concreting. Raheem et al. [59] suggested 10% CCA replacement as optimum for the production of interlocking paving stones. The addition of CCA as a pozzolanic material in blended cement shows a longer setting time than PC. They could therefore be most applicable when a low rate of heat development in mass concrete (low heat cement) is desired [22]. If the CCA SMC has a replacement rate of 20%, the substitution results in 0.8 Mtonnes PC, given a production of 79 Mtonnes and a 5% ash conversion rate.

### 5.4.3. Rice Husk Ash (RHA)

The rice husk or hull is the outermost hard protective covering layer that covers grains of rice, which is separated during the milling process [60]. Globally, 80 million tonnes of rice are produced annually which yields 16 million husks [26,53]. Because of its high silica content, rice husk ash is considered to be as an excellent pozzolanic material. The rice husk needs to be incinerated at a controlled temperature in order to obtain an optimum level of ash reactivity. The silica in the husk may become crystalline in too high or too low temperatures [22]. A temperature of 700 °C is normally reported to be an optimum temperature to calcine rice husk ash [25]. However, incineration temperatures of 900 °C have also been studied to produce acceptably reactive ash. Pulverizing or grinding this burnt ash increases the surface area, which further enhances its pozzolanic reactions when used as a binder [27]. The effect of varying replacement amounts of rice husk ash with PC has been reviewed previously [24]. In most cases, it has been observed that the use of rice husk ash can partially replace PC. When it comes to application of this material in block production, a 5% substitution of PC has resulted in a 6 days compressive strength of 0.363 MPa [60]. The total rice production in Africa is about 33 million tonnes/year [44]. The husk content is about 22% and the resulting RHA is about 5% of total crop mass [28]. This means that 1 tonne of rice gives 50 kg RHA. The total SSA RHA produced would be, based on this estimate, 1.65 million tonnes per year. Using the 20% PC replacement assumption, the yearly potential PC substitution becomes 330,000 tonnes.

5.4.4. Volcanic Ash (VA)

Volcanic ash or tuff deposits are abundant in Eastern Africa, along the rift valleys. Volcanic ash is used for cement grinding in large quantities both in Kenya and Uganda. In Uganda, most of the pozzolans are derived from vast volcanic ash deposits found in the west and southwestern areas of Bushenyi, Kabarole, Kasese, Bundibugyo, Kabale and Kisoro. Pozzolan deposits also occur in the eastern parts of the country on the slopes of Mt. Elgon and in Karamoja. Kampunzu and Mohr [61] put the quantities of pozzolan in the western rift at 100,000 cubic kilometres. Given a national annual demand of 3 million tonnes of cement in Uganda, one would need 600 years to deplete just one cubic kilometre. Although no known studies give specific statistical details on the abundance, it is agreed that the deposits are substantial.

At the moment, two cement companies (La Farge/Hima Cement Limited and Tororo Cement Ltd., Uganda) are actively mining the volcanic material and using it as cementitious mineral admixtures in their products marketed as Portland-pozzolan cement [62]. According to the mining data from the Uganda Bureau of Statistics, Statistical abstract 2020, 686,564 tonnes were produced in 2015; 846,604 tonnes in 2016; 792,564 tonnes in 2017; 1,103,198 tonnes in 2018 and 960,363 tonnes in 2019 [63]. These huge production volumes, and from only two cement producing companies, are an indicator that the pozzolan deposits in Uganda are quite sizeable [64]. Studies carried out on the pozzola deposits [62,65] have found volcanic ash to perform well and meet the requirements of ASTM C618, which is the Standard Specification for Coal Fly Ash and Raw or Calcined Natural Pozzolan for Use in Concrete. The compressive strength analyses in the same studies show that cement replacement of 20–25% with pozzolan can be applied in the production of cement, while replacement of 30–40% gives compressive strengths above the minimum required for masonry work and other non-structural purposes.

Although accurate statistical data are not available, geological information indicates availability of volcanic ash/pumice/scoria with substantial amount of pozzolan in Tanzania [66,67]. Volcanic ash sourced in the Rift Valley in the southwestern part of the country is currently used as a natural pozzolan substitute of imported fly ash by Mbeya Cement Company. The accessible VA of sufficient quality is high but difficult to assess correctly. In the summary in Table 9, we have assessed the total available amount to at least 1000 Mtonnes.

**Table 9.** Estimated amounts of available SCMs and potential to substitute PC in SSA. The cement substitution performance is a conservative assessment.

| Supplementary Cementitious Material in SSA | Amounts | Estimated Cement Substitution Performance | PC That Could Be Substituted | Comments |
|---|---|---|---|---|
| Volcanic Ash | >1000 Mtonnes | 20% | >200 Mt | Mainly East Africa but also in Cameroon |
| Calcined Clay | >1000 MtonVs | 20% | >200 Mt | Broad availability |
| Rich Husk Ash | 1.65 Mtonnes/year | 20% | 0.3 Mt/year | Production is widely spread |
| Cassava Peel Ash | 13.8 Mtonnes/year | 20% | 1.4 Mt/year | Production is widely spread |
| Corn Cob Ash | 3.95 Mtonnes/year | 20% | 0.8 Mt/year | Production is widely spread |

5.4.5. Calcined Clay (CC)

Reducing the clinker content in cement by replacing it with calcined clay and limestone has been suggested as a viable option for lowering cement costs and lowering $CO_2$ emissions. Limestone Calcined Clay Cement (LC3) is a novel blended cement made by combining limestone, calcined clay, clinker, and gypsum in predetermined proportions [42].

The low clinker percentage and the wide availability of raw materials, such as clays and limestone, make this kind of cement potentially economical in countries such as Kenya [41]. LC3 is also environmentally sustainable, as it can efficiently reduce $CO_2$ emissions by 30–50%. Furthermore, the manufacture of LC3 cement does not necessitate large capital investments because existing cement plants may be modified and adapted to the purpose. However, the LC3 production is tied to cement plants and therefore may not represent a valid alternative for local production in rural areas. However, one possibility is that calcined clays could be used together with locally produced burnt lime. The mix needs to be ground and some equipment investments are needed. A furnace for burning clay at 600–800 °C and to burn limestone at 1000 °C could be constructed using local materials. The product then needs to be ground, which could be achieved by small and relatively cheap ball mills produced in India and China. A small production plant, based on what could be seen as appropriate technology, and the use of relatively low-cost labour could be competitive for producing SCMs for block production. Investments would be needed but these could be in the range of 0.1–1% of a new cement plant estimated at about USD 100–200 M and would need a substantially smaller market area than cement plants of 0.5–1 million tonnes per year.

Primary, hydrothermal, residual, mixed, and secondary geneses of African kaolin clays have been identified in SSA. Sedimentary kaolins accounted for 50% of the reported kaolins, whereas primary kaolins accounted for 35%. According to [2], the main kaolin deposits were found in Central and Southern regions of Africa. Hydrothermal kaolins were primarily found in Central and West Africa. Kaolins from North Africa were mostly secondary clays.

Secondary kaolins of sedimentary origin were also found in large deposits and occurrences in Southern and West Africa [68]. Although kaolin has been discovered in large quantities in various sub-Saharan African countries, its industrial use is limited. The majority of kaolin deposits and occurrences are easy to reach and can be treated and beneficiated. Kaolins in mineable proportions have also been discovered in Precambrian and Permian clayey deposits.

Calcined clay has been utilized as an efficient pozzolan in mortar and concrete for many years due to its pozzolanic reactivity and high amount of silica and alumina. Previous studies have shown that substituting calcined clays for PC in mortar and concrete reduces $CO_2$ emissions, improves concrete durability, and delays the onset of age-related strength development [69]. The advantages of calcined clay cement content, which reduced calcium hydroxide in concrete and transformed it to C-S-H and C-A-S-H, have been linked to their pozzolanic reactivity.

Two studies have achieved a 100% replacement of PC using calcined clay and bio-based SCMs. According to Thiedeitz et al. [70], a mixture of brick preparation using 200 g RHA, 600 g of calcined clay and 50g of water yielded a brick of 5 MPa. Raheem et al. [71] proved that a mixture of corn cob ash and calcined clay yielded a brick with a 4.49 MPa compressive strength.

This summary does not provide sufficient information of available quantities or of the potential performance that calcined clays could have when used together with slaked lime or PC for concrete or for blocks. The accessible calcined clay is high but difficult to assess correctly. In the summary in Table 9, we have assessed the total available amount to at least 1000 Mtonnes.

*5.5. Potential for Substituting PC with Pozzolanic Materials in Block Production*

In this section we assess the theoretical improvement potential of PC substitution for the main materials identified. Table 9 is based on data obtained for the quality and quantity of the chosen pozzolanic materials in Table 6 and conclusions in Table 7.

The yearly production of the studied agricultural ashes totals 2.5 million tonnes, which compared to the estimated amount of cement used for blocks is about 4%. Compared to agricultural ash substituting potential, using volcanic ash and calcined clay seems to have

a larger potential based on the large available quantities. However, these are not renewable materials but still could be of use over several years. Table 10 summarises the sustainability opportunities for the selected SCMs for block-based housing construction in SSA.

**Table 10.** Estimated maximal theoretical sustainability improvement potential for cost saving, reduction of $CO_2$ emissions and employment creation, based on SCM substitution of PC for blocks in residential housing.

| SCM | PC That Could Be Substituted Mt/Year | Potential Value M USD/Year (USD 150/Tonne of PC) | Carbon Dioxide Emission Savings Mt/Year | Employment Created (in Man Years) |
|---|---|---|---|---|
| Volcanic Ash (1%/year) | 2 | 300 | 1 | 33,000 |
| Calcined Clay (1%/year) | 2 | 300 | 1 | 33,000 |
| Rich Husk Ash | 0.3 | 45 | 0.15 | 5000 |
| Cassava Peel Ash | 1.4 | 210 | 0.7 | 23,000 |
| Corn Cob Ash | 0.8 | 120 | 0.4 | 23,000 |
| **Total** | **6.5** | **980** | **3.3** | **About 100,000** |

Calcined clays and agricultural ashes would generate some carbon footprint but much lower than cement clinker that is burnt at 1450 °C. Moreover, about 60% of the cement clinker $CO_2$ emissions originate from the main raw material limestone. Possibly bioenergy could be used, which would mean that the only footprint generated is from limestone calcining. The savings are the difference between 700 kg $CO_2$/tonne cement and the ones generated by SCM production. The assessed difference here is set at 500 kg $CO_2$/tonne cement. The yearly employments created have been based on the assumption that every tonne of SCM produced enables the conversion of 20% of the potential value into employment based on a yearly salary of USD 1800.

In Table 10, the assumption has been made that yearly 1% of the available ash and clay deposits could be used. This results in that, yearly, 2Mt of PC could be substituted by volcanic ash and calcined clay each. The total yearly maximal theoretical PC substitution in SSA could amount to 6.5 million tonnes, with a potential value of about USD 1 billion per year. In order to assess the magnitude for employment creating it has been assumed that out of the potential value created, 20% would be used for creating employment. This means that the yearly savings should be reduced by 20% with the potential value becoming about USD 0.8 billion per year. Yearly carbons savings could be up to 3.3 Mt. Using the potential SCMs could create about 100,000 yearly jobs.

It is important to note that this is an assumption based on that all available material could be used. A rule of thumb used is that about 50% of the theoretical potential can be realised. Using this and rounding up the figures results in that yearly about 3.3 Mt of PC can be substituted to a value of USD 400 Million. $CO_2$-emissions could be reduced with about 1.7 Mt. About 50,000 jobs could be created.

Based on a yearly consumption of PC in SSA of about 150 Mt the SCMs could replace 2%. In block production where the estimate is that some 63 Mtonnes of cement are used for blocks in walls (see Table 8), SCMs could substitute about 5%. This indicates that even if SCMs seem to constitute an interesting option for PC substitution, other solutions are needed to secure production of sustainable blocks.

## 6. Discussion and Implications

The gaps in reported information on the use of SCMs in SSA block production derived from the scoping review in Table 7 indicate a low level of measurement maturity when compared with proposed needs. This could of course be the result of a too narrow search,

but at the same time, the ten identified reviews were selected among the top 300 search results. Indicating that if the norm of framing SCM reviews were to include the information listed in the checklist (Section 2.2), which should have surfaced among some of the ten reviews.

Based on a series of assumptions it has been possible to present an indication of the sustainability improvement potential for SCM-use in residential building blocks. Important assumptions are:

- It will be possible to use 50% of the existing SCMs.
- The SCMs will be able to replace 20% PC.
- The quantities of volcanic ash and calcined clays used per year could be up to 2 Mtonnes per year each.
- The average price of cement is USD 150/tonne.
- Each tonne of PC substituted by SCMs saves 500 kg $CO_2$.
- Out of the SCM value created 20% could be converted into new jobs based on a yearly salary of USD 1800.
- Standard requirements have not been considered since these vary between countries and since it should be possible to modify standards if block durability can be established at lower compressive strength.

A further implicit assumption is that the value created by using SCMs as PC substitution is sufficient to pay for investments needed in equipment. This is an area remaining for future research. The economic viability of each business case for using SCMs will depend on the size of the local market and the availability of raw materials.

The diagnosing performed represents a novel approach in the framing of SCMs as part of a sustainability opportunity. Despite all assumptions, it still is possible to highlight the magnitude of the opportunity for substituting PC in block production with SCMs. The improvement potential is both important and insufficient. It is important because of potential economic value creation and possible carbon reductions. It is insufficient as a solution for significantly substituting PC. There are several examples in current literature expressing a view that SCMs can and will contribute to a more sustainable use of resources and lesser emissions of $CO_2$ through substitution of PC, see for example the following quotes:

> " … the current state of the concrete industry is not sustainable. However, the utilization of industrial and agricultural waste components can be a breakthrough to make the industry more environmentally friendly and sustainable."— Shafigh et al. [26] p. 111

> "Over the past few decades, OPC usage in concrete has widely been criticized for its adverse environmental impacts associated with excessive limestone mining and high carbon dioxide ($CO_2$) emissions. … One alternative to reduce greenhouse gas emissions is to partially replace OPC with pozzolanic materials"— Thomas et al. [7] p. 1

The results of this study clearly indicate that SCMs could contribute, but that additionally other solutions are needed. Since affordability is important, this means that Carbon Capture and Storage is problematic as a solution due to its high cost. Other solutions for low cost and low carbon binders need to be developed.

There seems to be a gap in understanding between how SCMs are tested and where they could be used. Testing is performed using standard mortar testing based on EN 196-1 and testing the proposed binders in ordinary concrete. This implies applications where standard cement would be delivered by an ordinary cement plant. Cement plants are built to be used with fixed raw materials often in large quantities that are supplied regularly. For most agricultural ashes and probably also for calcined clays these demands would be impossible to meet. Volcanic ash is used as cement raw material in countries like Kenya and Uganda. This lowers production costs and the carbon footprint of cement but does not make cement cheap. Relying on cement plans would limit the locations and the potential cost saving since transport costs still would be high. From a cement plant perspective,

the business model for using SCMs works with volcanic ash, which is available in large quantities in one location. For other materials there would be little business sense. It could even be that low-cost binders for block making are only perceived as competition by the cement industry.

For materials like cassava peel ash to be relevant it should be possible to process this in a smaller industrial plant delivering binders to block production. This means that it should be possible to establish a value chain starting from raw material preparation, going over binder production to block production and then transportation for use in a residential building. It should be possible to achieve this without any support from a cement plant. Portland cement still could be used as a raw material, but all processing would be conducted in a separate company. For this to work there needs to be appropriate and affordable technology for calcining and grinding. Without a viable business model that is manageable for local entrepreneurs the potential of many of the pozzolanic materials cannot be realised. Current research does not seem to cover this part.

For testing to be relevant, it should be performed with the typical mixes used for blocks, which could be 5–7% cement and 93–95% sand for a sandcrete block. This means that the standard mortar test EN 196-1 needs to be adapted for this level. The cement content by weight is 22% in the EN 196-1 with a water to cement ratio of 0.5. For correctly comparing a PC and an alternative binder mix they need to be tested with the typical w/c used in blocks which are well above 0.5. Based on preliminary testing performed at Makerere University, a test mix should be about 150 g cement, 1800 g sand and 225 g water. This means a w/c ratio of 1.5, cement content of 7% and a water to material ratio similar to the one in EN 196-1 testing. The water to material ratio is important for compaction. Reducing the water added will make the mix harder to compact. Reduced compaction will lead to lower bulk density and lower strength.

SCMs and reference cement should be compared at the level of 5–7% binder. All examples of testing found in current research are with standard mortar only where the binder in the EN 196-1 testing standard is 22% binder. Possible substitution rates based on testing with standard mortar and testing with higher w/c content could differ substantially.

Issues, such as feasibility of collection of materials and further processing by calcining and grinding, need to be understood. Financial and technological viability, including alternative use-cases, need to be understood for realistic assessments of existing opportunities.

The resulting identified opportunity, derived from the diagnosing part of the Sustainability Opportunity Study (SOS) [11], is based on limited information. This is not a major problem since the purpose was to establish the magnitude of improvement potential.

The next step for work with SCMs in blocks is to test the assumption of 20% substitution of PC. Testing needs to be performed at the strength level that blocks are used. The possible rate of substitution and the cost of producing the SCMs will make it possible to assess business cases small binder production units. An important part of future research is understanding when there is a business case. Future research also needs to look into standard issues. The best option for innovation would be to focus on standards that focus on block performance where requirements are set based on the common areas of use.

## 7. Conclusions

Based on the results from the scoping review we conclude that there is a gap in the connection between the increasingly acute need for $CO_2$-neutral and affordable housing in SSA, and the aggregated research that could enable such solutions. Affordability and $CO_2$ neutrality are among the most important sustainability aspects for the studied context. Therefore, given that this is only the part of diagnosing in an opportunity study—assessing the improvement potential—it makes sense to start by scanning available options based on these two impacts. Then, for those options that seem promising (i.e., where there seems to be a significant opportunity in relation to those two aspects), more in-depth studies should be conducted that apply a systems perspective and check that these options still are smart

stepping stones towards full sustainability. This approach will help us avoiding dead-ends that lead to sub-optimization or other emerging problems.

For the case of sustainable housing solutions in SSA, which arguably is one of the most urgent challenges for SMCs to contribute to, the scoping review results in an inadequate pool of information. The ten review articles are complementary to each other for several of the requested information points; however, no review is complete, per definition of the information checklist developed for the case of SCMs for sustainable housing. With a predominant focus on chemical composition and with a mixed reporting of usability, solely focused on high performance concrete applications, the reviews do not provide sufficient information for accurately assessing SCM improvement potential in cement-based blocks (see Table 7).

With a combination of information from the scoping review and other sources, the resulting estimate for sustainability opportunities for block-based housing solutions in SSA indicate that there is substantial potential in using agricultural residues, such as rice husk ash, cassava peel ash and corn cob ash, but this potential is dwarfed by the potential of calcined clays and volcanic ash. The studied SCMs are estimated to have the potential to substitute about 3 Mtonnes of Portland cement consumption per year in SSA, which is about 5% of the total consumption for PC in blocks. This in turn corresponds to some 1.7 Mtonnes of savings in $CO_2$ emissions, and USD 400 million of value creation going from the cement industry to for more affordable housing. Some 50,000 yearly jobs could be created.

Going forward, the continued research on bio-based SCMs could have significant impact on the local level, where availability of PC and SCMs will be critical for the sustainability opportunities. However, for research efforts seeking solutions for SSA, there seems to be most potential in calcined clays and volcanic ash deposits. Continued research on low performance binder mixes, such as [70,71], contributing to the aggregated knowledge production for SCMs for sustainable housing solutions are identified as key future research areas.

**Author Contributions:** Validation, A.B. (Arezou Babaahmadi), A.B. (Apollo Buregyeya), K.O., S.R., A.R. and L.V.; Investigation, M.R., J.M.M. and K.O.; Writing—original draft, R.I.; Project administration, A.H. All authors have read and agreed to the published version of the manuscript.

**Funding:** This research has been supported by several grants, which we are very grateful for. These grants are: Vinnova grant for the project: Circular and climate neutral cement industry with enforced application of alternative cementitious materials in construction: Need for Standardization, market analysis and policy making (SCM-Force). Formas grant for the project: Crucial transition to application of cement replacement material: A heuristic approach to identify drivers and barriers, and to propose potential solutions (SCM-Force-II). Formas networking grant: Alternative binders for blocks used in residential building in Sub Saharan Africa—opportunities for reduced climate footprint, lower costs and increased employment. VR networking grant: Affordable and low carbon building in Sub Saharan Africa.

**Conflicts of Interest:** The authors declare no conflict of interest.

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
