# Peer review of "Supplementary Cementitious Materials in Building Blocks—Diagnosing Opportunities in Sub-Saharan Africa"

_sustainability, doi:10.3390/su15075822_

Round 1

Reviewer 1 Report

Diagnosing Sustainability Opportunities for Alternative Block 2 Binders in Sub Saharan Africa

This is an interesting paper on a worthwhile topic.  Having said that, it is the responsibility of reviewers to identify areas that could be improved.  Generally, feedback inevitably seems negative - and few comments encourage authors.  Please note that the comments I have made are intended as constructive criticism - and are not intended as a points-scoring exercise.

Please review the post-its associated with the highlighted text.

With respect to the method used for RQ1, the authors acknowledge that their search might have overlooked some texts.  This is not surprising because the search terms noted (line 241 and 242) are porous. This is a weakness of this paper…  Other researchers should be able to replicate the results obtained and this will be challenging in this case.

There is contradiction about fly ash, ground granulated blast furnace slag and silica fume (line 281 and 282).  These materials are noted to be in short supply and were therefore not considered - but then later in the paper its is stated that they have indeed been considered.

Considerable heat (and therefore CO2) needs to be expended to produce most of the SCMs...  This is acknowledged towards the end of the paper but not quantified.  Ideally these data should inform (and ideally be included in) Table 9.

Author Response

Thank you very much for your thorough work. . 

In the attached file we have added feedback to the reviewer.

Best regards

Reviewer 2 Report

General Comment: 

This paper, titled "Diagnosing Sustainability Opportunities for Alternative Block Binders in in Sub Saharan Africa ", reviews SCMs and their potential to promote sustainability in SSC. While the study looks interesting, the paper needs to be significantly improved before considering for publication. First of all, the manuscript requires proofreading to fix many grammatical issues (i.e., missing punctuations, choice of words, etc.) and improve the language. Moreover, this paper does not qualify to be a research article. It could be considered as a review article once improved. I list below some critical comments that need to be considered:

Comments about the abstract:

·      Line 21: Please correct SMC to SCMs. 

·      Lines 22 to 26 need to be clarified. Consider re-writing them to convey the idea in a simple way. 

·      Line 27: Please use "approximately" instead of "some" to improve scientific writing. 

Comments about the title :

·      The preposition “in” is mentioned twice. Please correct it.

·      I would suggest replacing “diagnosing” with “investigating.”

·      Please correct “Sub Saharan” to Sub-Saharan”

Comments about section 1, "Introduction":

·      Line 42: There are two periods (..) at the end of the sentence. Please correct it. 

·      Line 45: Do you mean 150 kg per person per year NOT “and year”? If this is the case, you can say the annual consumption is 150 kg per person.  

·      Line 58 to 59: “alternatives seem not to be used in ordinary concrete production or in cement block production to any greater extent.” SCMs have been extensively studied in the literature, and it is being used in concrete. For example, the use of pozzolanic Portland cement in concrete. It may not be used widely in cement blocks, but the claim that they are not being used in ordinary concrete production may not be accurate. Please correct this statement or add a reference to support it. 

·      Line 63: “major alternatives for PC substitution” PC can never be totally substituted. It can be partially replaced with SCMs. Please correct this statement. 

·      Line 72: “or the entire fraction of PC in specific applications” Could you please explain how the bonding mechanism will happen if you substitute the entire fraction of PC? As far as I know, you will still need PC since pozzolans react with the hydration products resulting from PC hydration to form bonding materials. Please note that these materials are Supplementary Cementitious Materials, NOT Cementitious materials. 

·      Line 75: This paper is more of a review paper since it is just examining the opportunities for SCMs. The research element needs to be improved, and the novelty should have been highlighted or mentioned.  

·      Line 76: “We focus” better to use passive voice rather than active voice. 

·      Line 77: The same comment as the previous one. 

·      Line 84: Please explain again how you will substitute all PC. 

·      Line 95: again, assessing available information is more of a review paper. This manuscript provides a good review for SCMs to be used in SSC. 

·      Please add the basic components of the cement blocks used and their proportions. Is there any standard or code? If so, please mention it.

·      Please add some features for SSC. For example, area, population, number of countries, etc. 

Comments about section 2, "Theoretical background”:

Consider removing this section and adding it to the introduction. It is a literature review. 

Line 110: SCMs are not binding materials. They need to be used with Cement to form binding materials. 

Comment about section 2.1 , “ Sustainability Opportunity Study for SCMs in SSA”:

The entire section is descriptive and does not add value to your manuscript. Moreover, consider constructing a schematic diagram to replace some parts of the text, table 1, and table 2.

Comment about section 2.2 , “ Theory of use of SCMs”:

Again, this is a literature review. Consider adding them to the introduction. 

Comment about section 5:

This section and sub-sections have good information in terms of a literature review. Again, the paper could be restructured to become a good review article.  

Best wishes to all authors

Author Response

Thank you very much for your comments.

In the attached file we have added feedback to the reviewer.

Best regards

Reviewer 3 Report

I want to thank you for being able to evaluate this exciting work entitled "Diagnosing Sustainability Opportunities for Alternative Block Binders in Sub Saharan Africa".

The authors are to be congratulated for presenting an exciting work that addresses the issue of sustainability in the production of cementitious materials in the Sub-Saharan Africa (SSA) region.

The objective of the manuscript was to evaluate the improvement potential of Supplementary Cementitious Materials (SCMs) in the production of block-based materials. The results show that the aggregated information about the hydraulic potential of different SCMs and the available quantities are limited. This work evaluates the theoretical improvement potential for replacing cement in ash-based blocks of other calcined materials.

In general, the authors' approach is quite interesting and very striking because it presents in numbers the gains that can be obtained with the use of SCMs.

Below I leave my comments on the topics of the manuscript:

In general, it would be interesting to review the manuscript as a whole for minor English and also typographical errors.

For example, in the title of the manuscript “Binders in in Sub Saharan Africa”

In general, the work presents a large number of tables; it would be interesting to assess whether all of these are necessary or if graphs or figures could replace some.

Summary: The summary is clear and succinct, my only consideration would be to replace "Sub-Saharan" Africa (SSA) instead of Sub-Saharan Africa (SSA).

Introduction: In general, the Introduction is very clear and perfectly contextualizes the importance and motivation of the proposed work.

On line 34 UN-Habitat instead of UN Habitat.

On line 42, ".." correct punctuation

In line 47, Cement-based blocks instead of "Cement based blocks" it is considered a change to do the same for the others.

On line 60, "is given by [6]" put a comma after the quote.

On line 83, "are included, It " Correct punctuation.

On line 84, "stabilization".

On line 91, "minimize".

2. Theoretical background: Very well-reasoned and clear.

On line 137, when the authors mention the difficulty of defining and measuring sustainability, it would be very interesting if the authors presented some proposals from the literature and presented the consideration for this work.

Table 1. I believe this table does not add much to work, but if the authors think it is essential, it would be interesting to represent this information through an infographic.

On line 164, "and cooling The" correct the punctuation.

3. Methodology and data: Congratulations again to the authors as they present a very clear and complete section.

On line 249, "Supplementary Cement tious Materials" typographical error in cementious.

4. Results from scoping review on SCMs: The main points raised on this topic are very clear, bringing exciting questions about the gaps found in the studies studied.

5. Sustainability opportunities for block production in SSA: Excellent topic, very clear, and with valuable information.

Regarding Figure 1, it would be interesting to improve the aesthetics (color changes, placing the respective units on the axes and not just on the legends) of the graph presented to make it clearer and more striking, I believe that this would be able to value this excellent work even more .

On line 337, "about 57 million1 tonnes" what does the superscript mean?

5.4.2 Corn Cob Ash; Abbreviations of et al. -fix several are without the period.

Conclusion: Another clear and excellent topic of this work.

Overall, the work is very good, and I believe it may be considered for publication in this journal.

Author Response

Thank you very much for your feedback and encouraging comments.

In the attached file I have added feedback to the reviewer.

Best regards

Round 2

Reviewer 2 Report

I highly appreciate the tremendous efforts spent by the authors to improve the paper. However, the manuscript still needs to be proofread by a professional editor to improve scientific writing. 

Line 73: Please avoid using a bold statement such as "there is no information..". As far as I know, these materials have been used in concrete mixes, and many of them have been reported in the literature. 

Although the authors consider this manuscript a research article, it looks more like a review article and does not qualify as a research article. I recommend this manuscript for publication as a review article. I leave it to the Journal's editor for the final decision.

Best wishes to all authors.

Author Response

Thanks for your persistence. We have responded to the best of our understanding.

Authors
